∂ | **Open Peer Review** | Clinical Microbiology | Research Article

# Urine PCR testing as an effective method for early diagnosis of leptospirosis

Paul Le Turnier,[1,2] Mathilde Zenou,[1] Mathieu Nacher,[2] Nicolas Higel,[3] Jean Jaonasoa,[4] Jean-François Carod,[5] Alexis Fremery,[6] Thomas Blanchot,[7] Pascale Bourhy,[8] Mona Saout,[9] Loïc Epelboin,[1,2] Mathieu Picardeau[8]

**ABSTRACT**  The role of urine PCR for leptospirosis diagnosis in the first week is controversial due to assumed limited urine excretion. This study analyzed the prescribing practices and sensitivity of blood and urine Leptospira PCR, particularly in early stages of infection. A study was conducted on adult patients diagnosed with leptospirosis in French Guiana between 2016 and 2022 by positive Leptospira PCR, or titer of Micro-Agglutination Test > 1/200, or positive IgM with no alternative diagnosis. The timing of PCR tests and their sensitivity were analyzed. A multivariate logistic regression was performed to identify the factors associated with the sensitivity and predict the probability of having a positive result. Among 188 analyzed patients, 137 (73%) and 61 (32%) underwent blood and urine PCR tests with a median (IQR) delay since symptoms onset of 5 (3–7) and 6 (5–8) days, respectively. The overall sensitivity of urine PCR was 84% (vs 70% for blood PCR, *P* = 0.04). Of the 25 patients sampled the same day in the first week, eight had negative blood PCR but positive urine PCR. Contrary to urine, the sensitivity of blood PCR significantly decreased with time since symptom onset (aOR 0.56 per day, 95% CI [0.44–0.73]). The predicted probability of positive urine PCR appeared higher than that of blood PCR as soon as 4 days after symptom onset. Urine PCR should be considered at the first consultation, alongside blood PCR.

**IMPORTANCE**  The study investigates the utility of urinary PCR for diagnosing leptospirosis, focusing on its early sensitivity. Conducted in French Guiana between 2016 and 2022, the research analyzed prescribing practices and the sensitivity of blood and urine PCR tests in adult patients. Results indicated that while underutilized, urinary PCR demonstrated high sensitivity from the early days of symptom onset, achieving 84% sensitivity compared to 70% for blood PCR. The sensitivity of blood PCR decreased over time, whereas that of urinary PCR remained stable. The study advocates for the routine inclusion of urinary PCR in the diagnostic workup from the first week of illness to enhance early leptospirosis detection.

**KEYWORDS**  leptospirosis, diagnosis, PCR, urine

L eptospirosis is a worldwide zoonotic disease caused by pathogenic *Leptospira* spp. with nonspecific clinical presentations in most cases (1). Biological diagnosis is, therefore, mandatory to confirm the clinical suspicion, enable antibiotic treatment, and provide an accurate estimate of the disease burden, especially in endemic areas. The diagnosis of leptospirosis is currently based on a combination of three different tests with specific limitations depending notably on the timing of sampling (2). Diagnosis can be made by the detection of anti-*Leptospira* immunoglobulin M (IgM) antibodies which appear 5–6 days after the onset of symptoms and persist for months, using a commercial kit or an in-house technique using an enzyme-linked immunosorbent assay (ELISA) or other technique such as a lateral flow immunoassay. The second test is the

**Peer Reviewers** Scott Nabity, Massachusetts General Hospital, Harvard Medical School, Boston, Massachusetts, USA; Reza Banihashemi, Razi Vaccine Serum Institute, Karaj, Alborz, Iran

Address correspondence to Paul Le Turnier, paul.leturnier@gmail.com.

The authors declare no conflict of interest.

microscopic agglutination test (MAT), which allows the detection of specific antibodies to the infecting serogroup starting 1 week after onset of symptoms and persisting over a year post-symptom. The MAT has long been considered the gold standard, but this test is performed by few reference laboratories, and many limitations exist, notably its poor performance in early samples (3). Lastly, DNA from pathogenic leptospires can be detected in blood, cerebrospinal fluid, or urine of patients through real-time PCR, targeting one of several specific markers. It is important to note that blood and urine cultures are not used for diagnosis as the culture of these slow-growing bacteria can take more than a month to yield positive results. The current gold standard for diagnosis of leptospirosis is debated because of limited performances, and the association of multiple diagnostic tests is deemed necessary to have the highest chance of obtaining a diagnosis of leptospirosis (2–4). Therefore, when faced with a patient with a clinical suspicion of leptospirosis, physicians are advised to consider the time elapsed since the symptom's onset (mainly fever) before prescribing the most appropriate tests for diagnosing the disease. Most patients with leptospirosis consult during the first week after symptom onset (5–7). Those with mild forms may be managed on an outpatient basis after a single blood test. Among them, those who recover quickly may decide to skip follow-up consultations, while others may not be further evaluated or tested because of the distance from care facilities and transportation issues (8). Thus, all efforts should be made to confirm the diagnosis as soon as possible, especially on the day of the first consultation, to start the antibiotic therapy early and potentially improve the patient's outcome. It is thus interesting to consider the currently recommended tests during the first week of the disease. These tests are PCR testing in blood or serum as soon as the patient is symptomatic (septicemic phase) and IgM testing after at least 5 days of symptoms to search for an early elevated titer, or a seroconversion or a titer elevation—which are only detectable if samples are repeated. MAT is usually recommended after at least 10 days of symptoms; however, early samples can be valuable when combined with a repeat sample that reveals seroconversion or a significant increase in antibody titers.

Urine PCR testing during the first week is much more controversial. In fact, it is generally accepted that PCR testing for leptospirosis in urine is not relevant during the first week of illness—an assumption that also applies to urine culture. This assertion rests on the dynamics of the pathophysiology of leptospirosis, with an early phase of bacteremia preceding the spread to many organs, particularly the kidneys (1). Hence, urine PCR testing is generally recommended after the first week of illness, once the bacteria are present in the kidneys and are expected to start spreading in the urine. This position is reflected in numerous documents: the 2003 World Health Organization guidelines do not even mention urine PCR testing (9); the CDC suggests testing in urine collected after the first week of illness (10, 11); the NIH guidelines suggest that urine PCR should be performed after 5 days of illness (12). The French National Authority for Health guidelines suggest a delay of 15 to 25 days after the onset of symptoms before searching for the bacteria in urine (13). However, the French National Reference Center currently recommends performing PCR testing of both blood and urine samples in the first days of the disease (14). There hence seems to be an inconsistency at the highest levels about what should actually be done to diagnose leptospirosis. Such contradictions in the recommendations may lead to different diagnostic practices by clinicians in the field. We, therefore, aimed to assess the practice of blood and urine PCR testing in patients diagnosed with leptospirosis in French Guiana between 2016 and 2022 and to determine the sensitivity of urine PCR testing compared with blood testing, particularly at an early stage of the disease.

## MATERIALS AND METHODS

### Study design

This study is an ancillary study on the epidemiology of leptospirosis in French Guiana. We conducted a retrospective, multicentric observational study in the three main hospitals of French Guiana. Adult patients (age at diagnosis >15 years) diagnosed with leptospirosis and treated in Cayenne, Kourou, St. Laurent du Maroni, and the French Guianese remote health centers between 1st January 2016 and 31 December 2022 were included.

We collected the population demographics, date of consultation, start of antibiotics, and various biological diagnostic tests and occurrence of severe forms. A severe form of leptospirosis was defined by the presence of any of the following criteria: use of vasopressor agents for hemodynamic support, mechanical ventilation, extrarenal epuration, or death.

### Biological diagnosis

During the study period, the PCR used for diagnosis was an in-house real-time PCR targeting the 16S RNA coding gene and *lipL32* gene. PCR in urine was as available to prescribers as PCR in the blood.

To perform the PCR, 200 µL of plasma, serum, or whole blood was extracted on MagNA Pure 96. Then, 10 µL of DNA was then added to the "Applied Biosystems Master Mix for universal TaqMan PCR" from Fisher Scientific Master Mix containing two primer pairs. The first primer pair targets the RNA16S gene (15), while the second pair targets the gene encoding the major outer membrane protein LipL32 (16) and the internal control primer pair (Phage M13). The two primer pairs targeting the leptospirosis genes are encoded by a 5′-FAM/3′-TAMRA probe, while the internal control primer pair is encoded by a 5′-VIC/3′-TAMRA probe. Amplification was then performed on ABI7500 or QS5 thermal cyclers, amplifying for 45 cycles. Urine PCR was carried out using the same protocol, although extraction was preceded by a 30 min centrifugation step at 12,000 rpm of 2 mL of the urine sample. Part of the supernatant (1.5 mL) was then discarded, leaving 0.5 mL of urine with the pellet.

Serological diagnosis of IgM was performed with an ELISA (Serion, Germany) in the Eurofins Biomnis private laboratory in France. Only positive tests were considered inclusion criteria, and equivocal tests were considered negative because of the limited specificity. When IgM titers were specified in the laboratory database, only titers above the 50 IU/mL significance threshold were defined as positive. The microscopic agglutination test (MAT) with 11 antigens was performed in routine practice by Biomnis lab. The serovars tested in the panel were as follows: Icterohaemorrhagiae, Copenhageni, Canicola, Australis, Pyrogenes, Pomona, Autumnalis, Grippotyphosa, Castellonis, Sejroe, and Patoc.

### Case definitions and inclusion and exclusion criteria

The diagnosis of leptospirosis was confirmed if there was either a positive PCR in a biologic sample and/or a MAT-positive titer of 1/400 or more for one of the pathogenic serovar. The diagnosis of leptospirosis was considered probable in the case of a MAT-positive titer of 1/200. Patients with positive anti-*Leptospira* IgM above 50 IU/mL alone (without positive MAT or PCR) were considered probable cases only if no differential diagnosis was found in the medical records, whether seroconversion of IgM occurred or not. Thus, patients with positive IgM antibodies alone (without MAT or PCR criteria) and with a differential diagnosis were excluded. Other exclusion criteria were as follows: missing patient files, opposition expressed by the patient to participate in the study, or age <18 years at the time of the data collection.

### Statistical analysis

For this study, we considered that having any of the criteria (see below) of either confirmed or probable leptospirosis case was sufficient for the biological diagnosis

of leptospirosis. The positivity of blood or urine PCR among the study patients was therefore considered here as a measure of its sensitivity.

Quantitative variables were described with medians and interquartile ranges, and qualitative variables were described with frequencies and percentages. Exact Fisher tests or Chi² tests were used to compare frequencies, as appropriate. The McNemar test was used to compare the sensitivity of blood and urine PCR tests for patients who had underwent both tests.

For the analysis, we only considered the first sample of urine or blood collected for PCR testing.

Only the first collected samples were considered for the analysis of sensitivity of blood and urine PCR tests.

To assess the association between the sensitivity of blood and urine PCR and the delay elapsed between symptom onset and sampling, a linear regression line was fitted, and the correlation was assessed using Spearman's rank correlation coefficient (rho).

Logistic regression was used to identify factors associated with positive blood and urine PCR test results and to estimate crude and adjusted odds ratios (aORs) and 95% confidence intervals (CIs) for the associations between exposure variables and PCR test results. The following variables were included in the model: male gender, age (+1-year increment), severe form, time between the onset of symptoms and biological fluid sampling (+1-day increment), and antibiotic intake prior to sampling. Variables with *P* values < 0.20 in the univariate analysis were used in the multivariable logistic regression model.

Antibiotic intake prior to sampling was defined as having been administered at least the day before blood or urine sampling for PCR testing.

Finally, the adjusted predicted probabilities of positive PCR based on this multivariable logistic regression model were estimated using the *margins* command in STATA. This command was employed to compute the marginal effects, which provide an interpretation of the relationship between the independent variables and the probability of a positive PCR result.

The *P* value significance level was set at 0.05. All tests were two-sided.

Statistical analysis was performed using STATA 15.1 (StataCorp LP, College Station, TX, USA). Graphics were obtained with EXCEL and STATA.

## RESULTS

### Characteristics of blood and urine PCR tests

A hundred and eighty-eight patients were included: 138 (73.4%) were confirmed cases with either positive PCR or MAT ≥1/400 cases, and 50 (26.6%) were probable cases with positive IgM without an alternative diagnosis or MAT ≥1/200. At least one blood or urine sample was analyzed for *Leptospira* PCR testing in 143 (76.1%) patients: 137 (73%) had blood PCR testing and 61 (32.4%) had urine PCR testing (see Fig. 1).

Among the 55/143 (38.5%) patients who were tested with both blood and urine PCR tests, 30/55 (54.6%) were sampled on the same day for both samples, with 25/30 (83%) being collected during the first week.

Table 1 presents the population characteristics and the timing of the PCR tests according to the samples collected for PCR testing.

Antibiotic therapy prior to sampling was twice as frequent for urine PCR than blood PCR (70.5% vs 33.3%, *P* < 0.001).

Urine PCR was performed less frequently in women than men (19.6% vs 36.6%, *P* = 0.032); no gender difference was noted for blood PCR (76.1% vs 71.8%, *P* = 0.573).

Figure 2 shows the number of tests (right axis) according to the delay between symptom onset and sampling. The minimum delay between symptom onset and blood or urine sampling was 0 and 2 days, respectively.

Most patients were sampled during the first week of illness, with 115/136 (84.6%, one missing data) and 41/61 (67.2%) for blood and urine PCR tests, respectively. Most patients were not sampled for blood PCR analyses on the day of consultation, but on

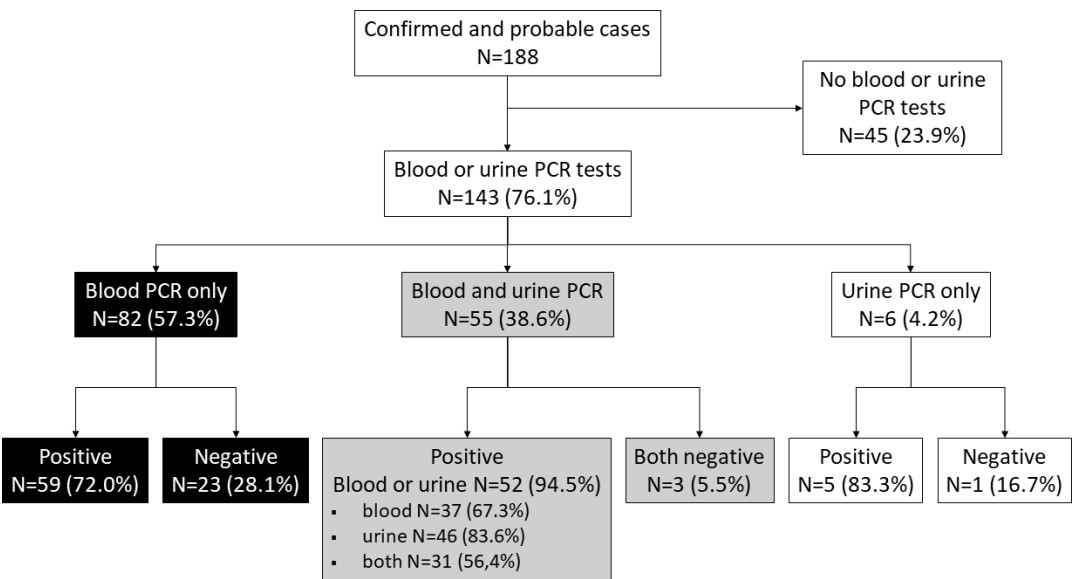

**FIG 1** Flow chart of distribution and results of blood and urine PCR tests in the study patients.

the next day or the day after, an even more pronounced trend for urine analysis was observed (see Table 1).

## Sensitivity of blood and urine PCR tests

Among the 143 patients with leptospirosis who were tested with blood or urine PCR test, the overall sensitivity of PCR was 81.1% (CI95 [0.74–0.87]) (116/143). Blood and urine PCR had a sensitivity of 70.1% (CI95 [0.62–0.78]) (96/137) and 83.6% (CI95 [0.72–0.92]) (51/61), respectively, regardless of the time elapsed since the onset of symptoms. During the first week, blood and urine PCR had a sensitivity of 78.3% (90/115) and 87.8% (36/41), respectively ($P = 0.183$).

Figure 2 shows the sensitivity (left axis) according to the delay between symptom onset and sampling.

In the 55 patients who had undergone both tests, the overall sensitivity of pooled PCR was 94.5% (CI95 [0.85–0.99]) (52/55) (see Fig. 1). Among them, the sensitivity of blood and urine PCR was 67.3% (CI95 [0.53–.79]) (37/55) and 83.6% (CI95 [0.71–0.92]) (46/55), respectively ($P = 0.049$).

Table 2 shows the PCR results of 30 patients with concomitant (sampled the same day) blood and urine samples. A third of these patients had a negative blood PCR and a positive urine PCR, including 7/16 (43.8%) patients sampled between 5 and 7 days after symptom onset.

Table 3 presents the details of all leptospirosis diagnostic tests performed for the 36 patients with positive urine PCR during the first week of illness. Of these, 6/36 (16.7%) had no other diagnostic criteria: blood PCR was negative (5/6) or not done (1/6), IgM test was negative (3/6) or not done (3/6), and MAT was not done (6/6).

## Factors associated with blood and urine PCR sensitivity

Linear regression lines with Spearman's correlation tests are presented in Fig. 3. Blood PCR sensitivity was highly correlated with the delay elapsed since symptom onset (Rho = −0.976, $P < 0.001$) contrary to the urine PCR sensitivity (Rho = −0.006, $P = 0.98$).

The logistic regression analysis of factors associated with blood and urine PCR test sensitivity is presented in Table 4.

In the crude analysis, antibiotic intake >1 day before blood sampling and a longer delay since the onset of symptoms before sampling were significantly associated with

TABLE 1 Population characteristics and leptospirosis diagnostic test results according to the PCR testing practices[c,d]

| Variable | All (n = 143) | Blood PCR only (n = 82) | Urine and blood PCR (n = 55) | Urine PCR only (n = 6) |
|---|---|---|---|---|
| Age, years | 38 (30–53) | 35 (27–49) | 43 (31–58) | 43.5 (32–48) |
| Male gender | 108 (75.5) | 56 (68.3) | 46 (83.6) | 6 (100) |
| Outcome | | | | |
| Severe form | 26 (18.2) | 9 (11.0) | 17 (30.9) | 0 (0) |
| Death | 4 (2.8) | 2 (2.4) | 2 (3.6) | 0 (0) |
| Delay between the onset of symptoms and days[a] | | | | |
| First consultation | 3 (2–5) | 3 (2–5) | 4 (2–5) | 5.5 (5–6) |
| Blood sampling for PCR | 5 (3–7) | 4 (3–6) | 5 (4–7) | –[e] |
| Urine sampling for PCR | 6 (5–8) | – | 6 (5–8) | 8 (7–9) |
| Antibiotic intake prior to sampling[b] | | | | |
| Blood PCR | 45/135 (33.3) | 21/80 (26.3) | 24 (43.6) | – |
| Urine PCR | 43/61 (70.5) | – | 39 (70.9) | 4 (66.7) |
| Positive PCR testing | | | | |
| Blood or urine PCR | 116 (81.1) | 59 (72.0) | 52 (94.5) | 5 (83.3) |
| Blood PCR | 96/137 (70.1) | 59 (72.0) | 37 (67.3) | – |
| Urine PCR | 51/61 (83.6) | – | 46 (83.6) | 5 (83.3) |
| Positive IgM testing | 87/136 (64.0) | 33/56 (59.0) | 22/40 (55) | 2/4 (50) |
| Positive MAT testing | 24/35 (69) | 6/11 (54.6) | 5/10 (50) | 0/4 (0) |

[a]One missing data for the date of blood PCR sampling.
[b]Antibiotic intake prior to sampling was defined as administered since at least the day before sampling of blood or urine for PCR testing, two missing data (One missing data for the date of blood PCR sampling and one missing data for antibiotic initiation date).
[c]Quantitative variables are expressed as median (25th percentile-75th percentile), qualitative variables are expressed as number (%), and n/N is given when data are missing or when IgM or MAT were not performed.
[d]IgM, Immunoglobulin M; MAT, microscopic agglutination test; PCR, polymerase chain reaction.
[e]"–", not applicable.

decreased sensitivity of blood PCR, while having a severe form of leptospirosis was associated with a higher sensitivity.

In multivariable analysis, the time elapsed since symptom onset was the only factor independently associated with blood PCR positivity (aOR 0.55 CI95 [0.43–0.71], $P < 0.001$) corresponding to a 1.8-fold reduction after each day elapsed since symptom onset.

No factor was significantly associated with urine PCR positivity.

The probabilities of having either a positive blood or urine PCR with multivariable models are shown in Fig. 4. Overall, urine PCR did not appear less contributive than blood PCR during the first week, except for the first 3 days of illness. However, after the 3rd day of symptomatic illness, urine PCR may contribute more than blood PCR to diagnosis.

## DISCUSSION

The aim of this study was to focus on the relevance of the samples to be used for PCR diagnosis of leptospirosis. Among the patients managed in French Guiana for leptospirosis during the 2016–2022 period, urine PCR was used half as often as blood PCR, but was mostly used in the first week. During this early phase, the sensitivity of urine PCR was higher than expected and appeared at least as contributive as blood PCR. The sensitivity of blood PCR quickly dropped during the first week, whereas the sensitivity of urine PCR was maintained high.

The lower use of urine PCR than of blood PCR in this study suggests that it may be perceived by clinicians as having a lower diagnostic yield compared to blood PCR or IgM serology and is therefore less frequently performed in routine. In one of the two main laboratories in France performing leptospirosis PCR for city laboratories or numerous hospitals between 01/01/2022 and 31/12/2024, four times as many patients were tested by blood PCR (n = 6,823) than by urine PCR (n = 1,798) (not published data Alexia Barbry, BIOMNIS, France).

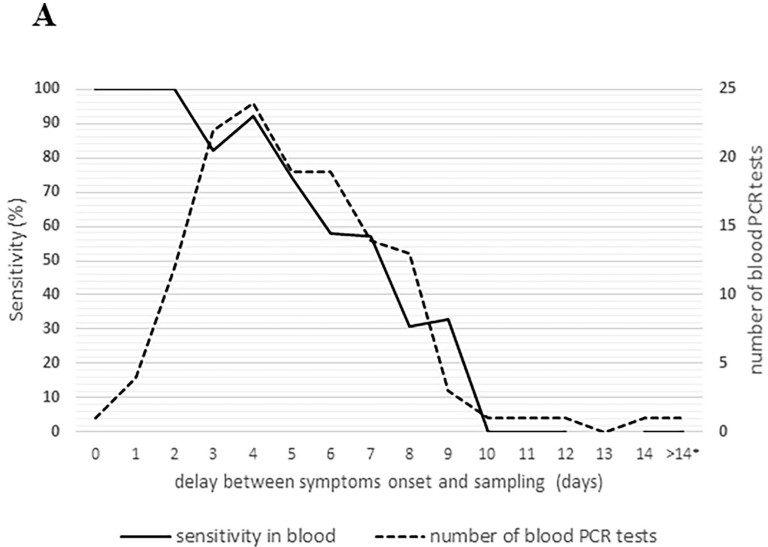

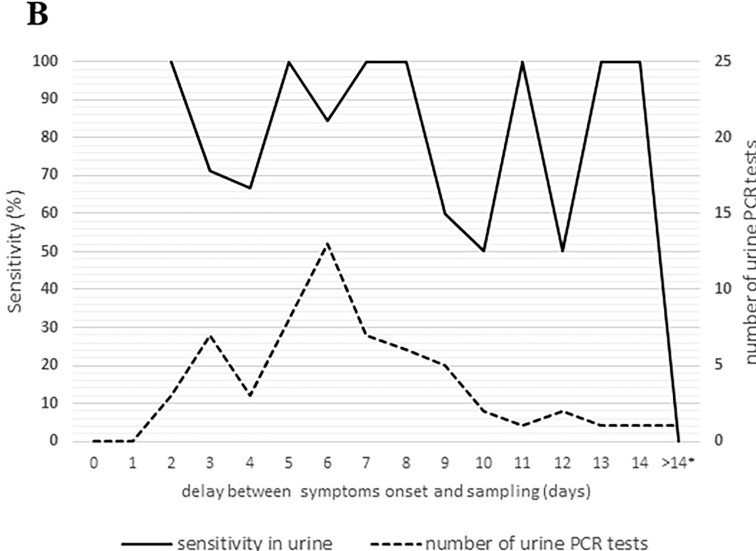

**FIG 2** Sensitivity of blood and urine PCR (left axis) and number of tests (right axis) according to the delay between symptom onset and sampling. (A) Blood PCR. * One patient had a blood sample collected 17 days after symptom onset. (B) Urine PCR. * One patient had a urine sample collected 26 days after symptom onset.

Surprisingly, most urine PCR tests were performed during the first week of illness in the present study. On the contrary, blood PCR tests were done relatively late. Indeed, the median delays since symptom onset and consultation, blood PCR sampling, and urine sampling were, respectively, 3, 5, and 6 days. This illustrates that blood and urine

**TABLE 2** PCR results in 30 patients with concomitant urine and blood sampling grouped by time elapsed since the onset of symptoms[b]

| Time elapsed since the onset of symptoms (days) | Positive blood PCR | Positive urine PCR | Both positive | Positive blood PCR and negative urine PCR | Negative blood PCR and positive urine PCR | P value[a] |
|---|---|---|---|---|---|---|
| 2–4 ($n = 9$) | 8/9 (88.9) | 7/9 (77.8) | 6 (66.7) | 2 (22.2) | 1 (11.1) | $P = 0.56$ |
| 5–7 ($n = 16$) | **9/16 (56.3)** | **16/16 (100)** | 9 (56.3) | 0 | 7 (43.8) | **$P = 0.0082$** |
| >7 ($n = 5$) | 3/5 (60) | 5/5 (100) | 3 (60.0) | 0 | 2 (40) | |

[a]The sensitivity of blood and urine PCR tests with McNemar test. Bold indicates a significant difference.
[b]Quantitative variables are expressed as median (25th percentile–75th percentile), and qualitative variables are expressed as number (%).

**TABLE 3** Details of diagnostic tests performed for the patients with positive urine PCR during the first week of illness ($n = 36$)[c]

| Patient | Blood PCR[a] | Delay[b] | Urine PCR[a] | Delay[b] | IgM[a] | Delay[b] | MAT[a] | Delay[b] |
|---|---|---|---|---|---|---|---|---|
| 1 | + | 2 | + | 2 | + | 8 | . | |
| 2 | + | 3 | + | 2 | . | | . | |
| 3 | + | 2 | + | 2 | . | | . | |
| 4 | + | 3 | + | 3 | − | 3 | . | |
| 5 | + | 3 | + | 3 | − | 3 | . | |
| 6 | + | 3 | + | 3 | . | | . | |
| 7 | + | 4 | + | 4 | − | 4 | . | |
| 8 | + | 3 | + | 4 | − | 3 | . | |
| 9 | + | 4 | + | 5 | . | | . | |
| 10 | + | 5 | + | 5 | + | 6 | . | |
| 11 | + | 5 | + | 5 | . | | . | |
| 12 | + | 4 | + | 5 | + | 5 | + | 11 |
| 13 | + | 5 | + | 5 | − | 5 | . | |
| 14 | + | 4 | + | 5 | − | 4 | . | |
| 15 | + | 4 | + | 5 | + | 4 | . | |
| 16 | + | 6 | + | 6 | − | 4 | − | 4 |
| 17 | + | 6 | + | 6 | . | | . | |
| 18 | + | 3 | + | 6 | . | | . | |
| 19 | + | 6 | + | 6 | + | 8 | . | |
| 20 | + | 5 | + | 6 | − | 4 | − | 4 |
| 21 | + | 6 | + | 6 | − | 6 | . | |
| 22 | + | 6 | + | 6 | + | 6 | . | |
| 23 | + | 7 | + | 7 | − | 7 | . | |
| 24 | + | 5 | + | 7 | . | | . | |
| 25 | . | | + | 3 | + | 7 | . | |
| 26 | . | | + | 7 | − | 6 | . | |
| 27 | − | 3 | + | 3 | + | 2 | . | |
| 28 | − | 5 | + | 5 | − | 4 | . | |
| 29 | − | 6 | + | 6 | + | 3 | . | |
| 30 | − | 5 | + | 6 | . | | . | |
| 31 | − | 6 | + | 6 | + | 4 | − | 5 |
| 32 | − | 6 | + | 6 | . | | . | |
| 33 | − | 7 | + | 7 | + | 7 | − | 2 |
| 34 | − | 7 | + | 7 | . | | + | 14 |
| 35 | − | 8 | + | 7 | . | | . | |
| 36 | − | 7 | + | 7 | − | 5 | . | |

[a]The diagnostic test results are presented as follows: positive (+), negative (−), and not performed (.).
[b]Delay between the onset of symptoms and the test mentioned in the previous column (days).
[c]IgM, Immunoglobulin M; MAT, microscopic agglutination test.

sampling for PCR tests were rarely done directly after physical examination, especially for urine PCR. This suggests that there is room for improvement regarding the time to perform PCR after consultation and to perform blood and urine PCR together.

Patients who benefited from both blood and urine PCR analysis had the best overall sensitivity, again highlighting the complementarity of these tests (3). Of note, patients with both tests performed regardless of the results were more frequently classified as severe. This could be explained by the fact that broader explorations were undertaken in these severe patients, including a more frequent urine PCR testing, rather than a higher level of positivity in severe patients.

In the present study, the urine PCR tests sampled during the first week of illness were often positive (87.8%), including for three urine samples collected after only 2 days of illness. Moreover, many patients with concomitant samples during the first week had

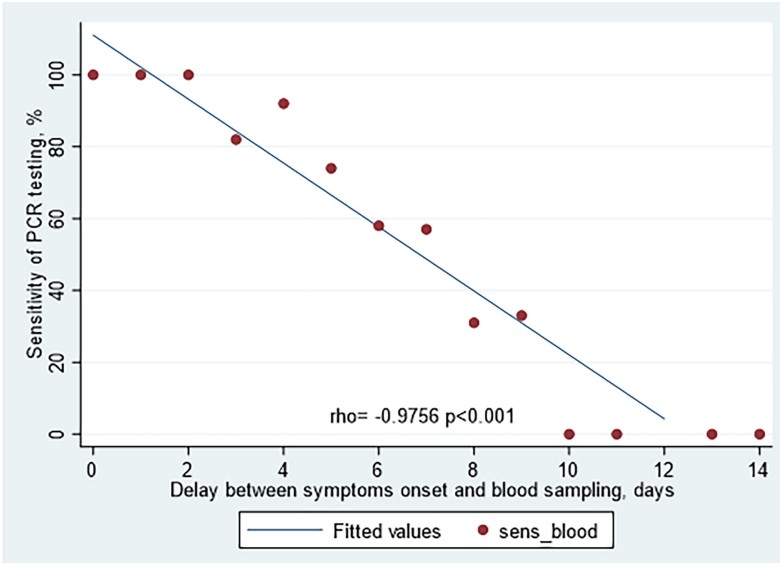

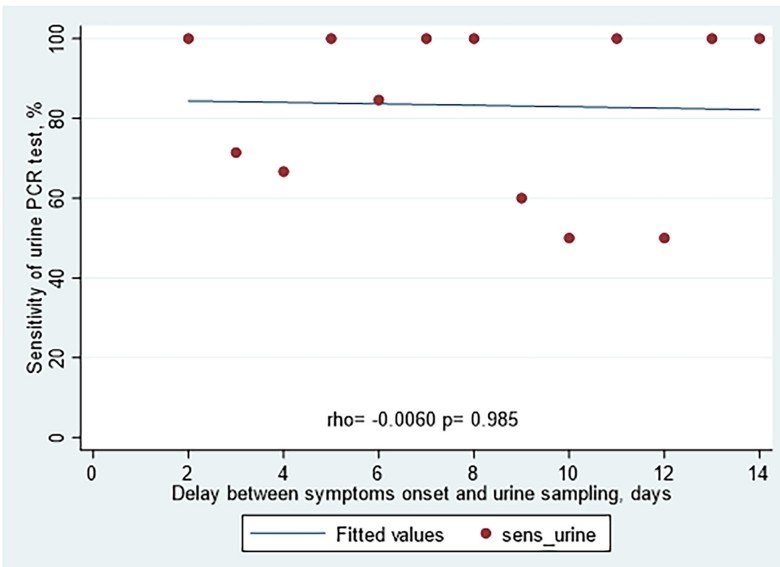

**FIG 3** Correlation between the sensitivity of PCR testing and delay between symptom onset and sampling during the first 14 days of illness. (A) Blood PCR. (B) Urine PCR.

discrepant results in favor of urine PCR, which is in contradiction with the presumed delayed urinary excretion.

So far, many studies that assessed the interest of urine PCR have not provided precise data on the timing of sampling, which is a major point to compare the respective contribution of tests to leptospirosis diagnosis (17–20). Another work did not specify the respective part of urine and blood PCR analyses in early diagnosis tests (21).

Woods et al. conducted a large prospective study in Laos using the same target genes as in the present study for diagnostic PCR testing (22). Using a Bayesian latent class modeling, they found that urine qPCR had a sensitivity of 45% (27.0–66.7) and a specificity of 99.6 (99.3–100). Of the 68 patients with positive urine PCR, more than half were sampled in the first week, and roughly a quarter were sampled before 3 days of

**TABLE 4** Logistic regression analysis of factors associated with positive blood and urine PCR results[e]

| Variable | Crude odds ratio | 95 CI | P value | Adjusted odds ratio | 95 CI | P value |
|---|---|---|---|---|---|---|
| Positive blood PCR[a] | | | | | | |
| Male gender | 0.92 | 0.393–2.135 | 0.838 | –[f] | – | – |
| Age [b] | 1.02 | 0.995–1.049 | 0.089 | 1.03 | 0.996–1.068 | 0.078 |
| Severe form | **3.99** | **1.125–14.145** | **0.014** | 4.04 | 0.892–18.304 | 0.070 |
| Time between symptoms onset and sampling[c] | **0.551** | **0.435–0.697** | **<0.001** | **0.554** | **0.430–0.714** | **<0.001** |
| Antibiotic intake prior to sampling[d] | **0.306** | **0.141–0.663** | **0.003** | 0.503 | 0.198–1.395 | 0.196 |
| Positive urine PCR[a] | | | | | | |
| Male gender | 0.597 | 0.066–5.387 | 0.646 | – | – | – |
| Age [b] | 0.993 | 0.9498–1.0378 | 0.750 | – | – | – |
| Severe form | 0.883 | 0.1998–3.901 | 0.869 | – | – | – |
| Time between symptom onset and sampling[c] | 0.879 | 0.746–1.034 | 0.120 | .8895 | 0.7491–1.0563 | 0.182 |
| Antibiotic intake prior to sampling[d] | 0.222 | 0.0259–1.901 | 0.170 | 0.2606 | 0.0297–2.2815 | 0.224 |

[a]Multivariable analysis was made for 135 patients (one missing data for PCR date and one missing data for antibiotic initiation date) for blood PCR and 61 patients (no missing data) for urine PCR.
[b]Age effect was analyzed with a 1-year increment.
[c]Time since symptom onset was analyzed with a 1-day increment.
[d]Antibiotic intake prior to sampling was defined as administered since at least the day before sampling.
[e]Bold indicates significant association (P value < 0.05).
[f]"–", not applicable.

illness, suggesting that urine PCR could be positive in the early phase and even on the first days of illness.

In a paper released 30 years ago, Bal et al. (23) reported the detectability of leptospires in early-collected urine samples. Of seven patients with urine and serum samples collected during the first 8 days of illness, all urine PCRs were positive, while only two serum PCRs were positive.

Warnasekara et al. reported a prospective hospital-based study, which included 1,734 clinically suspected leptospirosis cases and undifferentiated febrile patients. A minimum of 40% of cases would have been missed by using any of the following tests individually: whole blood qPCR, single or paired-sample MAT, urine qPCR, culture, culture qPCR, or surveillance case definition. The authors found that whole blood qPCR had a good sensitivity, including up to 15 days after symptom onset. They concluded that whole-blood qPCR should be the standard diagnostic test for leptospirosis during the first 10 days of illness in their context. Besides, the rate of positive urine PCR tests was very low with 20% of probable or confirmed at the first sample, which was a third of blood PCR. The authors suggested in the discussion section that the date of symptom onset may have been wrong for some patients, which could have biased the interpretation of delay. Furthermore, they insisted on the need to compile the various tests so as not to miss a diagnosis.

In the present study, for each day elapsed since symptom onset, the chance of having a positive PCR test in blood was divided by 1.8. The importance of an early testing for blood PCR has previously been reported and discussed (2). However, in contrast to blood PCR, there was no significant association between the time elapsed since symptom onset and the sensitivity of urine PCR. These results are concordant with the known pathophysiology of the disease that includes for some patients a delayed kidney tissue bacterial clearance (1, 24). These results are also in line with previous works that found a positive urine PCR sampled several months after the infection (23).

The graphical representation of predicted probabilities of having a positive blood or urine PCR over time (Fig. 4) provides insights regarding the relative contribution of these two tests to the diagnosis of leptospirosis. Up to the 4th day after the onset of symptoms, blood PCR appears to be more sensitive than urine PCR. After the 4th day, the ratio may be reversed. Considering that most patients consult between days 3 and 5, it would seem appropriate to recommend both tests and not to choose between blood or urine PCR testing. A clear message would be to propose the blood and urine test for any patient

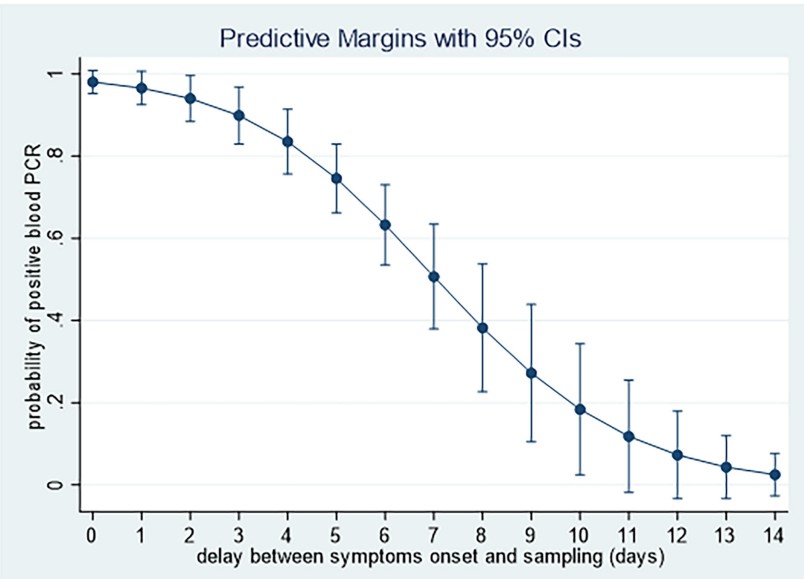

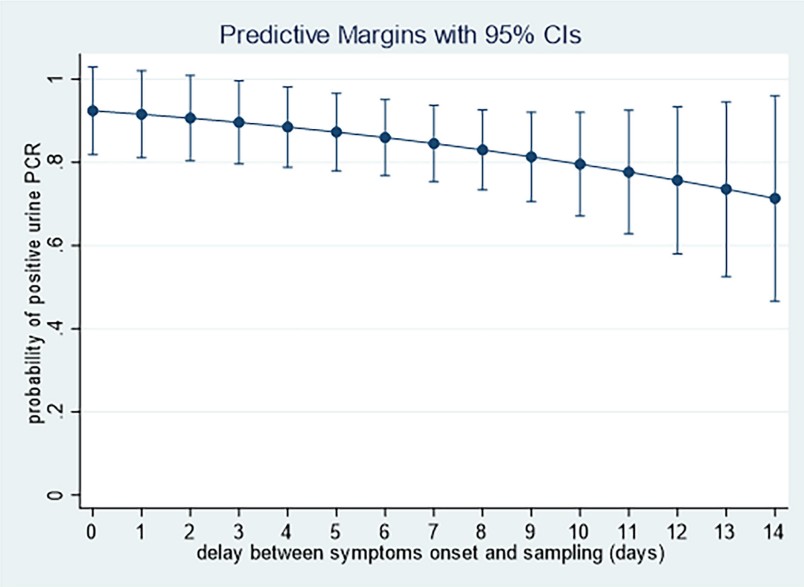

**FIG 4** Predicted probabilities of positive PCR based on the logistic regression multivariable models for blood PCR and urine PCR. (A) Blood PCR. (B) Urine PCR.

suspected of being in the acute phase of leptospirosis (1st week). Some authors already suggested a systematic approach and not to rely on the date of symptom onset for determining the choice of diagnostic tests including serology (25).

The analysis of the effect of antibiotic therapy initiation prior to sampling on PCR results was also relevant. First, antibiotic therapy was more frequently administered before sampling for urine PCR than for blood PCR, which could indicate a selection bias, as commented further in the study limitations. Furthermore, there was a negative association between antibiotic intake before sampling and blood PCR sensitivity in the univariate analysis, in line with previous statements on the negative impact of antibiotic intake on blood PCR (26). Nevertheless, after adjusting for time, this association was no

longer significant in the multivariable analysis. Interestingly, there was no association between antibiotic intake and urine sensitivity. Indeed, leptospires are highly sensitive to antibiotics, and their concentrations in blood may be lower than in urine depending on the stage of the disease, especially in non-severe cases. Thus, although antibiotic intake reduces the urinary excretion of live leptospires (27, 28), urine PCR may still be positive in patients who have been receiving antibiotics for several days.

The relatively low sensitivity of blood PCR in the present study (70%) and its lower sensitivity than urine PCR (84%) could be explained by several factors. First, using a higher volume of urine than blood for DNA extraction could have an impact. Second, at least a third of the patients were receiving antibiotics before blood sampling. Moreover, the delay between symptom onset and blood sampling for PCR analyses was relatively long with a median of 5 days considering the detection window of leptospires in blood is narrow for many reasons. For instance, patients with leptospirosis who are not receiving antibiotics when they are tested can have negative *Leptospira* blood PCR results in the last days of the first week because of the appearance of immune reactions that will control the infection. The limitations of blood PCR are particularly applicable in nonsevere forms where leptospiremia is low and limited in time (29), contrary to renal excretion.

Beyond the inherent limitations of the retrospective design, this study has several weaknesses. First, only patients with robust criteria for biological diagnosis of leptospirosis performed in routine were included. Some clinical guidelines consider an isolated positive IgM sufficient for a probable leptospirosis diagnosis. Yet, we acknowledged the potential lack of specificity of IgM, particularly due to cross-reactions that may introduce classification bias. To reduce false positives, patients with elevated IgM as the sole diagnostic criterion were excluded if an alternative diagnosis was present. Because the study included only patients with confirmed or probable leptospirosis, the specificity of urine and blood PCR tests was not ascertainable in our study. High sensitivity and specificity of the PCR tests used in this study have been reported. Over 20 years ago, Smythe et al. noted that some discrepancies between negative MAT and positive PCR actually reflected early diagnoses detected by PCR but missed by MAT, rather than true PCR false-positives. Although false-positives can occur with any PCR assay, they are rare with qPCR and are unlikely to account for the results observed. Finally, the absence of patients with other diagnoses than leptospirosis was not considered a major issue for the study objectives.

Second, the decision not to use the MAT as the gold standard can be criticized and appear as a limitation to the estimation of the sensitivity of PCR. However, not using MAT as a gold standard in the study was based on two major points. First, very few patients had samples tested with MAT, and even fewer had repeated tests over time (data not shown). Second, it is now admitted that MAT should be considered a gold standard for a retrospective diagnosis when repeated, which is rarely done in routine, as illustrated by the study presented here (4).

Third, the number of patients with concomitant blood and urine PCR was limited, as well as their distribution by day. Therefore, it was not possible to conduct the logistic regression analysis of the factors associated with positivity in this specific population. To corroborate the study findings, further work should focus on a larger subset of patients with concomitant blood and urine PCR testing during the first week.

Fourth, it is possible that bias was introduced when clinicians selected the type of clinical samples for leptospirosis diagnosis. This is suggested by the low proportion of negative PCR results among patients who underwent both blood and urine testing. In addition, the higher rate of antibiotic use before sampling for urine PCR than for blood PCR may also indicate clinical differences between patients who did or did not undergo urine testing. This may reflect a selection bias, with urine PCR performed preferentially in patients considered more likely to be infected.

Finally, the date of symptom onset was routinely gathered from symptomatic patients by physicians and retrospectively gathered from medical charts by investigators. As

a consequence, error in reporting the accurate date of symptom onset cannot be excluded, especially for patients who tend to minimize their initial symptoms, but this should be marginal (3). Besides, in routine practice, prescribers rely on information provided by the patient initially, so the pragmatic conclusions of this study remain relevant.

Although many factors can influence the detection of leptospires in urine by PCR analyses (30), progress has been made in the last decades (20, 22). Reshaping the optimal timing of urine PCR is challenging but relevant to optimize the diagnosis accuracy of leptospirosis since molecular tools are now widely available on the field including many resource-limited areas (30). Thirty years after the article by Bal et al. (23), it still seems necessary to acknowledge the contribution of urine PCR to the diagnosis of leptospirosis.

## Conclusion

This study highlights some advantages of urine PCR in supporting leptospirosis diagnosis during the first week of illness, contrary to prevailing assumptions. Although limited by the few concomitant blood and urine samples, our findings provide clinically relevant insights for optimizing diagnostic strategies. Additional prospective studies where sequential enrollees with febrile illness are tested with all referenced tests are needed to affirm the conclusions suggested by this study. Meanwhile, we suggest that urine PCR should be considered at the first consultation, alongside blood PCR.

### ACKNOWLEDGMENTS

The authors acknowledge Arsene Kpangon (Kourou General Hospital, French Guiana), Alexia Barbry (Biomnis laboratory, Lyon, France), Sabine Trombert Paolantoni (CERBA laboratory, Saint-Ouen-l'Aumône, France), and Hatem Kallel (Cayenne General Hospital, French Guiana) for their contribution to the data collection and their support for this study.

This research received no specific grant from any funding agency in the public, commercial, or not-for-profit sectors. P.L.T.'s position is supported by the Agence Nationale de Recherche sur le SIDA et les Maladies Infectieuses Emergentes (ANRS/MIE).

P.L.T.: Conceptualization, Formal analysis, Writing – original draft. M.Z.: Investigation, Writing – review & editing. M.N.: Formal analysis, Writing – review & editing. N.H.: Writing – review & editing. J.J.: Writing – review & editing. J.-F.C.: Writing – review & editing. A.F.: Writing – review & editing. T.B.: Investigation, Writing – review & editing. P.B.: Writing – review & editing. L.E.: Formal analysis, Writing – review & editing. M.P.: Writing – review & editing.

### AUTHOR AFFILIATIONS

[1]Department of Infectious and Tropical Diseases, Cayenne General Hospital, Cayenne, French Guiana

[2]CIC Inserm 1424, Centre Hospitalier de Cayenne, Amazonian Institute of Population Health, Cayenne, French Guiana

[3]Department of Intensive Care and Reanimation, Cayenne General Hospital, Cayenne, French Guiana

[4]Department of Microbiology, Cayenne General Hospital, Cayenne, French Guiana

[5]Department of Biology, St Laurent du Maroni General Hospital, St Laurent du Maroni, French Guiana

[6]Department of Emergency, Cayenne General Hospital, Cayenne, French Guiana

[7]Biomnislab Laboratory, Eurofins Biomnis, Lyon, France

[8]Biology of Spirochetes Unit, National Reference Center for Leptospirosis, WHO Collaborating Center for Reference and Research on Leptospirosis, Institut Pasteur, Paris, France

[9]Laboratoire Tropical Biome and Immuno Pathophysiology (TBIP), Université de Guyane, Cayenne, French Guiana

## AUTHOR ORCIDs

Paul Le Turnier  http://orcid.org/0000-0002-6164-311X
Mathieu Nacher  http://orcid.org/0000-0001-9397-3204
Alexis Fremery  http://orcid.org/0000-0001-5365-6191
Loïc Epelboin  http://orcid.org/0000-0002-3481-5991
Mathieu Picardeau  http://orcid.org/0000-0002-5338-5579

## AUTHOR CONTRIBUTIONS

Paul Le Turnier, Conceptualization, Formal analysis, Methodology | Mathilde Zenou, Investigation | Mathieu Nacher, Formal analysis | Thomas Blanchot, Investigation | Loïc Epelboin, Formal analysis, Supervision.

## ETHICS APPROVAL

The institutional review board at each site approved the protocol. Each participant was informed by letter of the possibility to oppose the utilization of his or her data in the study. The study was approved by the CERMIT Ethics Committee (number 2023-0702-2). Formal consent was not required for this type of observational retrospective study. All statistical analyses were performed on an anonymized data set, and procedures were in accordance with the ethical standards of the National Research Committee and the Declaration of Helsinki.

## ADDITIONAL FILES

The following material is available online.

### Open Peer Review

**PEER REVIEW HISTORY (review-history.pdf).** An accounting of the reviewer comments and feedback.

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
