## [Reviewer comments · Microbiology Spectrum]

Microbiology Spectrum

Urine PCR testing as an effective method for early diagnosis of leptospirosis

Paul Le Turnier, Mathilde Zenou, Mathieu Nacher, Nicolas Higel, Jean de la Croix Jaonaso, Jean Carod, Alexis Fremery, Thomas Blanchot, Pascale Bourhy, Mona Saout, Loic Epelboin, and Mathieu Picardeau

Corresponding Author(s): Paul Le Turnier, Centre Hospitalier de Cayenne

Review Timeline:

Submission Date:	April 15, 2025
Editorial Decision:	August 26, 2025
Revision Received:	September 5, 2025
Accepted:	September 8, 2025

Editor: Mark Pandori

Reviewer(s): Disclosure of reviewer identity is with reference to reviewer comments included in decision letter(s). The following individuals involved in review of your submission have agreed to reveal their identity: Scott Naby (Reviewer #2); Reza Banihashemi (Reviewer #3)

Transaction Report:

DOI: <https://doi.org/10.1128/spectrum.01185-25>

Re: Spectrum01185-25 (Urine PCR testing as an effective method for early diagnosis of leptospirosis)

Dear Dr. Paul Le Turnier:

Thank you for the privilege of reviewing your work. Below you will find my comments, instructions from the Spectrum editorial office, and the reviewer comments.

Revision Guidelines

Sincerely,
Mark Pandori
Editor
Microbiology Spectrum

Reviewer #2 (Comments for the Author):

This is a nicely executed analysis and summary of diagnostic results for sample persons with microbiologically confirmed leptospirosis at select clinical sites in French Guiana. It contributes interesting observational results regarding the diagnostic potential of urine PCR early in the course of leptospirosis.

The greatest limitation of the study is its cross-sectional acquisition of cases from hospital sites. Not all persons included in the

study received both blood and urine PCR, and the patients were not sampled systematically. There may have been bias introduced when clinicians chose to submit clinical diagnostic specimens (e.g. based on severity of clinical presentation, anuric status). Such bias may be indicated by the low percentage (5.5%) with negative blood/urine PCR in the group for which both samples were collected compared with either blood (28.1) or urine (16.7) alone. The frequency of empiric antibiotic use was twice that in the urine PCR group compared with the blood PCR group, perhaps further suggesting an important clinical difference between groups.

1. What is the frequency of false positive urine and blood PCR with this assay (compared with serological confirmation alone as gold standard)? It would be useful to see an alternative calculation of sensitivity for PCR compared with serologic confirmation (w/o PCR as part of the confirmed definition).

2. It must also be clearly stated as a limitation that only 35/143 (24.5%) of potentially eligible participants even had MAT testing -- which had traditionally been considered gold standard.

3. The first sentence in the Conclusion seems to miss what the study's data can reasonably be concluded to show. I'm not convinced that this is actually what your data show. They do show that urine PCR was less commonly used and that urine PCR appears to have some advantages in supporting leptospirosis diagnoses in the first week of illness. Additional prospective studies where sequential enrollees with febrile illness are tested with all referenced tests (notably both urine and blood PCR for all individuals) are needed to affirm the conclusions suggested by this study.

Reviewer #3 (Comments for the Author):

Major Strengths

- **Clinical Relevance:** The study addresses a challenging topic for clinicians and provides guidance that could lead to earlier diagnosis and, consequently, better patient outcomes.
- **Novel Findings:** The findings regarding the high sensitivity of urine PCR in the first week of illness are surprising and contradict some existing guidelines. These results directly emphasize the need to re-evaluate diagnostic protocols.
- **Robust Methodology:** The strong statistical analysis, utilizing multivariate logistic regression, correctly identifies factors associated with the sensitivity of blood PCR.
- **Concomitant Sampling:** The evaluation of combined sensitivity in patients who underwent both tests clearly demonstrates the complementarity of the two methods.

Key Points for Revision and Clarification

1. **Study Design:** In the "Methods" section, the authors use the term "cross-sectional study" but describe it as an "observational retrospective study" in the "Ethics" section. This needs clarification to avoid confusion. Many of the study's weaknesses are also due to its retrospective nature, which must be addressed more explicitly in the "Limitations" section.
2. **Case Definitions:** The diagnostic criteria for "probable cases," which are defined based on a positive IgM alone without a potential alternative diagnosis, may be open to criticism. Given the retrospective nature of the study, this is a strong assumption that should be discussed with more detail.
3. **Sampling Bias:** The authors note that urine PCR was prescribed less frequently than blood PCR. While this is an important finding, it could indicate a selection bias, meaning clinicians might have only ordered urine PCR for patients with specific symptoms or in a particular clinical phase. The potential for this bias should be discussed more thoroughly in the "Discussion" section.
4. **Influence of Antibiotics:** The study shows that antibiotic use before sampling was twice as frequent in the urine PCR group compared to the blood PCR group. While the authors included this factor in their multivariate analysis, this significant difference could confound the results and requires further discussion on its potential implications.
5. **Data Presentation and Transparency:** In the "Results" section, it is mentioned that among 30 patients with concomitant sampling in the first week, 8 had a negative blood PCR but a positive urine PCR. This is a very important finding, but the total number of concomitant samples and their distribution by day are limited, and this limitation should be clearly stated in the conclusion.

Final Recommendation

This manuscript has high potential for publication due to its clinical importance and novel findings. I recommend that the authors perform a Minor Revision. They must address the points raised above clearly and comprehensively, especially in the "Discussion" and "Limitations" sections, to strengthen the study's validity. After these revisions are made, the article would be well-suited for consideration for publication.

**Title**

Urine PCR testing as an effective method for early diagnosis of leptospirosis

**Authors:**

P Le Turnier^{1,2*}, M Zenou¹, M Nacher², N Higel³, J Jaonaso⁴, JF Carod⁵, A Fremery⁶, T Blanchot⁷, P
Bourhy⁸, Mona Saout⁹, L Epelboin^{1,2}, M Picardeau⁸

**Authors and affiliations:**

1: Department of Infectious and Tropical Diseases Cayenne General Hospital French Guiana.

2: CIC Inserm 1424, Amazonian Institute of Population Health, Centre Hospitalier de Cayenne, Cayenne
97300, French Guiana

3: Department of intensive care and reanimation Cayenne General Hospital French Guiana.

4 Department of microbiology Cayenne General Hospital French Guiana.

5: Department of biology, St Laurent du Maroni General Hospital French Guiana

6: Department of Emergency Cayenne General Hospital French Guiana.

7: Biomnislab laboratory Eurofins Biomnis Lyon France

8: Biology of Spirochetes Unit, National Reference Center for Leptospirosis, Institut Pasteur, 75015
Paris, France.

9: Laboratoire Tropical Biome and ImmunoPathophysiology (TBIP), Université de Guyane, Cayenne,
French Guiana;

**Key words**

Leptospirosis, diagnosis, PCR, urine

**Bullet points**

- • The optimal timing of urine PCR for leptospirosis diagnosis is controversial
- • Among 188 patients diagnosed with leptospirosis in French Guiana, urine PCR was performed
in half as many patients as blood PCR
- • Most urine PCR tests were done in the first week of illness and had a high positivity rate
- • In the first week, many patients had positive urine PCR and negative blood PCR

- • A longer delay between symptoms onset and sampling was associated with negative results for
blood PCR but not for urine PCR

***Corresponding author** (P Le Turnier)

Unité de Maladies Infectieuses et Tropicales, Centre Hospitalier de Cayenne, Centre Hospitalier
Universitaire Guyane 3 avenue Alexis Blaise, 97300 Cayenne, French Guiana

Phone +33 6 58 24 58 27; +594 594 39 50 40

Email: paul.leturnier@gmail.

**ABSTRACT (max 250, current word count 247)**

**Introduction**

The role of urinary PCR (PCRu) for leptospirosis diagnosis in the first week is controversial due to
assumed limited urinary excretion. This study analyzed the prescribing practices and sensitivity of
blood and urine *Leptospira* PCR, particularly in early stages of infection.

**Methods**

A study was conducted on adult patients diagnosed with leptospirosis in French Guiana between 2016
and 2022 by positive *Leptospira* PCR, or micro-agglutination test >200, or positive IgM with no
alternative diagnosis. The timing of PCR tests and their sensitivity were analyzed. A multivariate
logistic regression was performed to identify the factors associated with the sensitivity and predict the
probability of having a positive result.

**Results**

Among 188 analyzed patients, 137 (73%) and 61 (32%) underwent blood and urine PCR tests
respectively with a median (IQR) delay since symptoms onset of 5 (3–7) and 6 (5–8) days
respectively. The overall sensitivity of urine PCR was 84% (vs 70% for blood PCR, $p=0.04$). Of the
25 patients sampled the same day in the first week, 8 had negative blood PCR but positive urine PCR.
Contrary to urine, the sensitivity of blood PCR significantly decreased with time since symptoms
onset (aOR 0.56 per day, 95% CI [0.44-0.73]). The predicted **probability** of positive urine PCR
appeared higher than blood PCR as soon as 4 days after symptoms onset.

**Conclusion**

This study highlights the underuse of urine PCR despite its high early sensitivity, advocating for its
broader use alongside blood PCR, including in the first week of illness.

**Importance (max 150, word count 105)**

[revised manuscript text omitted]

Patient	Blood PCR*	Delay**	Urine PCR*	Delay**	IgM*	Delay**	MAT*	Delay**
1	+	2	+	2	+	8	.	
2	+	3	+	2	.		.	
3	+	2	+	2	.		.	
4	+	3	+	3	-	3	.	
5	+	3	+	3	-	3	.	
6	+	3	+	3	.		.	
7	+	4	+	4	-	4	.	
8	+	3	+	4	-	3	.	
9	+	4	+	5	.		.	
10	+	5	+	5	+	6	.	
11	+	5	+	5	.		.	
12	+	4	+	5	+	5	+	11
13	+	5	+	5	-	5	.	
14	+	4	+	5	-	4	.	
15	+	4	+	5	+	4	.	
16	+	6	+	6	-	4	-	4
17	+	6	+	6	.		.	
18	+	3	+	6	.		.	
19	+	6	+	6	+	8	.	
20	+	5	+	6	-	4	-	4
21	+	6	+	6	-	6	.	
22	+	6	+	6	+	6	.	
23	+	7	+	7	-	7	.	
24	+	5	+	7	.		.	
25	.		+	3	+	7	.	
26	.		+	7	-	6	.	
27	-	3	+	3	+	2	.	
28	-	5	+	5	-	4	.	
29	-	6	+	6	+	3	.	
30	-	5	+	6	.		.	
31	-	6	+	6	+	4	-	5
32	-	6	+	6	.		.	
33	-	7	+	7	+	7	-	2
34	-	7	+	7	.		+	14
35	-	8	+	7	.		.	
36	-	7	+	7	-	5	.	

273
274
275
276

IgM Immunoglobulin M, MAT Microscopic agglutination test

* The diagnostic test results are presented as follows: positive (+), negative (-), not performed (.)

** Delay between the onset of symptoms and the test mentioned in the previous column (days)

*Factors associated with blood and urine PCR sensitivity*

Linear regression lines with spearman correlation tests are presented in **Figure 3**. Blood PCR
sensitivity was highly correlated with the delay elapsed since symptoms onset (Rho = -0.976, p<0.001)
contrary to the urine PCR sensitivity (Rho= -0.006, p=0.98).

**Figure 3. Correlation between sensitivity of PCR testing and delay between symptoms onset and**
**sampling during the 14 first days of illness**

**Fig 3A Blood PCR**

**Fig 3B Urine PCR**

The logistic regression analysis of factors associated with blood and urine PCR tests sensitivity is
 presented in **Table 4**.

**Table 4 Logistic regression analysis of factors associated with positive blood and urine PCR**
 **results**

Variable	Crude Odds Ratio	95 CI	P value	Adjusted Odds Ratio	95 CI	P value
Positive blood PCR*						
Male gender	0.92	0.393-2.135	0.838	-	-	-
Age **	1.02	0.995-1.049	0.089	1.03	0.996-1.068	0.078
Severe form	3.99	1.125-14.145	0.014	4.04	0.892-18.304	0.070
Time between symptoms onset and sampling ***	0.551	0.435-0.697	<0.001	0.554	0.430-0.714	<0.001
Antibiotic intake prior to sampling****	0.306	0.141-0.663	0.003	0.503	0.198- 1.395	0.196
Positive urine PCR*						
Male gender	0.597	0.066 - 5.387	0.646	-	-	-
Age **	0.993	0.9498 - 1.0378	0.750	-	-	-
Severe form	0.883	0.1998 - 3.901	0.869	-	-	-
Time between symptoms onset and sampling ***	0.879	0.746 - 1.034	0.120	.8895	0.7491-1.0563	0.182
Antibiotic intake prior to sampling****	0.222	0.0259 - 1.901	0.170	0.2606	0.0297-2.2815	0.224

Bold indicates significant association (p value< 0.05)

* Multivariable analysis was made for 135 patients (1 missing data for PCR date and 1 missing data
 for antibiotic initiation date) for blood PCR and 61 patients (no missing data) for urine PCR

**age effect was analyzed with a one-year increment

*** time since symptoms onset was analyzed with a one-day increment

[revised manuscript text omitted]

We could also suggest that the initiation of antibiotic therapy could guide the choice of diagnostic
tests given the negative impact of antibiotic intake on blood PCR (26). Indeed, there was a trend in
multivariable analysis for a negative association between antibiotic intake prior sampling and blood
PCR sensitivity. However, there was no association in univariate or multivariable analysis between
antibiotic intake and urine sensitivity.

Finally, the relatively low sensitivity of blood PCR in the present study (70%) and its lower
sensitivity than urine PCR (84%) could be explained by several factors. First, using a higher volume of
urine than blood for DNA extraction could have an impact. Second, at least a third of the patients were
receiving antibiotic before blood sampling. Indeed, leptospires are very sensitive to antibiotic and
blood concentration of leptospires could be lower than urine concentrations, especially in non-severe
forms. Thus, even though antibiotic intake reduce the urinary excretion of live leptospires (27, 28),
urine PCR might still be positive in patients receiving antibiotic since several days.

Moreover, the delay between symptoms onset and blood sampling for PCR analyses was relatively
long with a median of 5 days considering the detection window of leptospires in blood is narrow for
many reasons. For instance, patients with leptospirosis who are not receiving antibiotic when they are
tested - can have negative *Leptospira* blood PCR result in the last days of the first week because of the
appearance of immune reaction that will control the infection. The limitations of blood PCR are
particularly applicable in non-severe forms where leptospiremia is low and limited in time (29)
contrary to the renal excretion.

This retrospective study has several limitations. First, only patients with robust criteria for biological
diagnosis of leptospirosis performed in routine were included. Therefore, the specificity of each
diagnostic test was not ascertainable in our study. However, urine PCR is known to harbor a high
degree of specificity (22). Thus, the absence of patients with other diagnosis than leptospirosis was not
considered as a major issue for the study objectives. Besides, using routine examinations allowed to
describe the practices and report that the diagnostic procedures were indeed heterogenous, as
illustrated by the lower urine PCR prescription rate for women with leptospirosis.

Secondly, the decision not to use the MAT as the gold standard can be criticized. It was based on two
points. First, very few patients had samples tested with MAT and even fewer had repeated tests over
time (data not shown). Second, it is now admitted that MAT can only be considered as a gold standard
to the diagnostic when it is repeated which is rarely done in routine (4).

Thirdly, the number of patients with concomitant blood and urine PCR was limited, which did not
allow to conduct the logistic regression analysis of the factors associated with the positivity in this
specific population. Further works should focus on this specific subset of patients. However, the main
conclusions drawn from this study – the high level of positive urine PCR in the first week and the

differential impact of time on blood and urine PCR sensitivity- should not be affected by this
limitation.

[revised manuscript text omitted]

- 16. Levett PN, Morey RE, Galloway RL, Turner DE, Steigerwalt AG, Mayer LW. 2005.
Detection of pathogenic leptospires by real-time quantitative PCR. *Journal of Medical Microbiology*
54:45–49.
- 17. Villumsen S, Pedersen R, Borre MB, Ahrens P, Jensen JS, Kroghfelt KA. 2012. Novel
TaqMan® PCR for detection of *Leptospira* species in urine and blood: pit-falls of in silico validation. *J*
*Microbiol Methods* 91:184–190.
- 18. Podgoršek D, Ružić-Sabljić E, Logar M, Pavlović A, Remec T, Baklan Z, Pal E, Cerar T.
2020. Evaluation of real-time PCR targeting the lipL32 gene for diagnosis of *Leptospira* infection.
*BMC Microbiology* 20:59.
- 19. Iwasaki H, Chagan-Yasutan H, Leano PSA, Koizumi N, Nakajima C, Taurustiati D, Hanan F,
Lacuesta TL, Ashino Y, Suzuki Y, Gloriani NG, Telan EFO, Hattori T. 2016. Combined antibody and
DNA detection for early diagnosis of leptospirosis after a disaster. *Diagnostic Microbiology and*
*Infectious Disease* 84:287–291.
- 20. Esteves LM, Bulhões SM, Branco CC, Carreira T, Vieira ML, Gomes-Solecki M, Mota-Vieira
522 L. 2018. Diagnosis of Human Leptospirosis in a Clinical Setting: Real-Time PCR High Resolution
Melting Analysis for Detection of *Leptospira* at the Onset of Disease. *Scientific Reports* 8:9213.
- 21. Fonseca C de A, Teixeira MMG, Romero EC, Tengan FM, Silva MV da, Shikanai-Yasuda
MA. 2006. *Leptospira* DNA detection for the diagnosis of human leptospirosis. *Journal of Infection*
52:15–22.
- 22. Woods K, Nic-Fhogartaigh C, Arnold C, Boutthasavong L, Phuklia W, Lim C, Chanthongthip
528 A, Tulsiani SM, Craig SB, Burns M-A, Weier SL, Davong V, Sihalath S, Limmathurotsakul D, Dance
D a. B, Shetty N, Zambon M, Newton PN, Dittrich S. 2018. A comparison of two molecular methods
for diagnosing leptospirosis from three different sample types in patients presenting with fever in
Laos. *Clin Microbiol Infect* 24:1017.e1-1017.e7.
- 23. Bal AE, Gravekamp C, Hartskeerl RA, De Meza-Brewster J, Korver H, Terpstra WJ. 1994.
Detection of leptospires in urine by PCR for early diagnosis of leptospirosis. *J Clin Microbiol*
32:1894–1898.
- 24. Sivasankari K, Shanmughapriya S, Natarajaseenivasan K. 2016. Leptospiral renal colonization
status in asymptomatic rural population of Tiruchirapalli district, Tamilnadu, India. *Pathog Glob*
*Health* 110:209–215.
- 25. Philip N, Affendy NB, Masri SN, Yuhana MY, Than LTL, Sekawi Z, Neela VK. 2020.
Combined PCR and MAT improves the early diagnosis of the biphasic illness leptospirosis. *PLoS One*
15:e0239069.
- 26. Picardeau M. 2013. Diagnosis and epidemiology of leptospirosis. *Med Mal Infect* 43:1–9.
- 27. Watt G, Padre LP, Tuazon ML, Calubaquib C, Santiago E, Ranoa CP, Laughlin LW. 1988.
Placebo-controlled trial of intravenous penicillin for severe and late leptospirosis. *Lancet* 1:433–435.
- 28. Edwards CN, Nicholson GD, Hassell TA, Everard CO, Callender J. 1988. Penicillin therapy in
icteric leptospirosis. *Am J Trop Med Hyg* 39:388–390.
- 29. Limothai U, Lumlertgul N, Sirivongrangson P, Kulvichit W, Tachaboon S, Dinhuizen J,
Chaisuriyong W, Peerapornratana S, Chirathaworn C, Praditpornsilpa K, Eiam-Ong S, Tungsanga K,
Srisawat N. 2021. The role of leptospiremia and specific immune response in severe leptospirosis. 1.
*Sci Rep* 11:14630.

30. Gunasegar S, Neela VK. 2021. Evaluation of diagnostic accuracy of loop-mediated isothermal
amplification method (LAMP) compared with polymerase chain reaction (PCR) for *Leptospira* spp. in
clinical samples: a systematic review and meta-analysis. Diagnostic Microbiology and Infectious
Disease 100:115369.

RESPONSE TO REVIEWERS (in bold, line numbering refers to the clean version of the revised manuscript)

Reviewer #2 (Comments for the Author):

This is a nicely executed analysis and summary of diagnostic results for sample persons with microbiologically confirmed leptospirosis at select clinical sites in French Guiana. It contributes interesting observational results regarding the diagnostic potential of urine PCR early in the course of leptospirosis.

The greatest limitation of the study is its cross-sectional acquisition of cases from hospital sites. Not all persons included in the study received both blood and urine PCR, and the patients were not sampled systematically. There may have been bias introduced when clinicians chose to submit clinical diagnostic specimens (e.g. based on severity of clinical presentation, anuric status). Such bias may be indicated by the low percentage (5.5%) with negative blood/urine PCR in the group for which both samples were collected compared with either blood (28.1) or urine (16.7) alone. The frequency of empiric antibiotic use was twice that in the urine PCR group compared with the blood PCR group, perhaps further suggesting an important clinical difference between groups.

We thank the reviewer for this comment. A bias in the type of test requested is entirely possible. Table 1 presents the detailed characteristics of the patients based on the type of PCR testing.

We can assume that testing for Leptospira PCR in urine is less acknowledged than in blood for non-specialists first line prescribers in emergency for example. PCR Urine testing might have been performed on patients who were considered more likely to have an infection, leading to an infectious disease consultation that may have guided the decision to order this test.

As the reviewer points out, the frequency of empirical antibiotic therapy prior to testing was indeed higher in patients tested by urinary PCR, and the severity of the patients was greater in the group that underwent both tests. However, although antibiotic therapy was more frequently administered prior to urinary PCR testing, this demonstrates that urinary PCR remains relevant (given its positivity rate) despite prior antibiotic therapy, which seemed to be less the case for blood PCR as demonstrated in bivariate analysis.

The discussion has been modified to raise this point **LINE 361-366:**

“Fourthly, it is possible that bias was introduced when clinicians selected the type of clinical samples for leptospirosis diagnosis. This is suggested by the low proportion of negative PCR results among patients who underwent both blood and urine testing. In addition, the higher rate of antibiotic use before sampling for urine PCR than for blood PCR may also indicate clinical differences between patients who did or did not undergo urine testing. This may reflect a selection bias, with urine PCR performed preferentially in patients considered more likely to be infected. ”

1. What is the frequency of false positive urine and blood PCR with this assay (compared with serological confirmation alone as gold standard)?

;

Our results were based on the work of Smythe et al. and Levett et al., which supported the choice of these diagnostic tests for our study.

In the study by Smythe et al. (BMC Infect Dis 2002), no true false positives were observed: the four samples that were initially PCR+/MAT- were in fact confirmed infections (positive by culture and/or later seroconversion). If serology (MAT) is taken as the sole gold standard, 4/49 ($\approx 8\%$) would appear as “false positives,” but these represent early diagnoses rather than true false positives. On the other hand, the sensitivity of PCR compared with MAT alone is very low (0% in this series), since none of the 17 MAT-positive cases were detected by PCR. Of note, in this study the serum samples used for PCR testing were not only acute sera of patients suspected of having leptospirosis but also patients with at-risk occupations which could explain such high disparity of positive MAT and negative PCR. In addition, in this paper, the MAT titer positivity threshold was 1:50, a threshold more relevant for seroprevalence studies than comparing PCR performance for acute leptospirosis infection.

In the study by Levett et al no false positive was reported neither. The study primarily assessed sensitivity based on *Leptospira* culture. Specificity was evaluated using cultures from 38 different *Leptospira* species, as well as bacteria and fungi. Notably, the LipL32 PCR demonstrated excellent performance, particularly in its ability to detect only pathogenic *Leptospira*. Human samples were limited to two urine specimens and one serum sample with a positive serology, although the authors did not provide detailed data regarding these human tests.

We summarized briefly this point in the discussion section **LINE 343-347**

“High sensitivity and specificity of the PCR tests used in this study have been reported. Over 20 years ago, Smythe et al. noted that some discrepancies between negative MAT and positive PCR actually reflected early diagnoses detected by PCR but missed by MAT, rather than true PCR false positives. Although false positives can occur with any PCR assay, they are rare with qPCR and are unlikely to account for the results observed.”

It would be useful to see an alternative calculation of sensitivity for PCR compared with serologic confirmation (w/o PCR as part of the confirmed definition).

This remark is entirely relevant if we consider that serological confirmation is the gold standard. As discussed in the article, serological confirmation by MAT has long been considered as the gold standard; It can be considered as a gold standard when performed appropriately: with late samples showing elevated titers or repeated samples and a rise in titer and using a relevant panel of serogroups for local diagnosis (see Waggoner et al. Curr Opin Infect Dis 2017). Many experts point out that PCR may be considered as the new gold standard for early diagnosis, given its performance in the initial management of patients, but that there are limitations to PCR and that it is probably necessary to consider performing all tests as the most effective strategy for maximizing the probability of making a diagnosis with biological confirmation (Warnasekara et al plos NTD 2022, Valente et al. BMC Inf Dise 2024, Agampodi AJTMH 2016). To this end, the Warnasekara article emphasizes how complementary the tests are. Similarly, another article in the manuscript points out that a relevant approach could ultimately be to request all tests from the outset, considering that the onset of symptoms is sometimes unclear, particularly at the time of initial care, and that it may be inappropriate to limit the request for certain tests based on this criterion.

To respond more directly to the point raised by the reviewer: it would indeed be useful to evaluate the sensitivity of PCR using MAT as a gold standard in our study. However, it is not feasible because of several reasons. There was a very limited number of patients with MAT and the number of patients with repeated MAT on convalescent serum was even more limited as mentioned as such in the discussion section **LINE 351-356 . Therefore, it does not seem relevant to apply a sensitivity analysis considering only patients with positive MAT tests as gold standard. Considering the intrinsic limitations in terms of sensitivity (PCR to some extent, MAT particularly in the early stages) and specificity (IgM), a combined approach would provide the most solid evidence as highlighted in the**

discussion and as suggested by many authors. This has already been highlighted in the discussion section see **LINE 308-313:**

“Considering that most patients consult between days 3 and 5, it would seem appropriate to recommend both tests and not to choose between blood or urine PCR testing. A clear message would be to propose the blood and urine test for any patient suspected of being in the acute phase of leptospirosis (1st week). Some authors already suggested a systematic approach and not to rely on the date of symptoms onset for determining the choice of diagnostic tests including serology (25).”

2. It must also be clearly stated as a limitation that only 35/143 (24.5%) of potentially eligible participants even had MAT testing --- which had traditionally been considered gold standard.

This statement was added to modify the discussion as follows:

Line 350-355

“Secondly, the decision not to use the MAT as the gold standard can be criticized and appear as a limitation to the estimation of the sensitivity of PCR. However, not using MAT as a gold standard in the study was based on two major points. First, very few patients had samples tested with MAT and even fewer had repeated tests over time (data not shown). Second, it is now admitted that MAT should be considered as a gold standard for a retrospective diagnosis when repeated, which is rarely done in routine as illustrated by study presented here.”

3. The first sentence in the Conclusion seems to miss what the study's data can reasonably be concluded to show. I'm not convinced that this is actually what your data show. They do show that urine PCR was less commonly used and that urine PCR appears to have some advantages in supporting leptospirosis diagnoses in the first week of illness. Additional prospective studies where sequential enrollees with febrile illness are tested with all referenced tests (notably both urine and blood PCR for all individuals) are needed to affirm the conclusions suggested by this study.

Thank you for this comment. Indeed, the first sentence of the conclusion highlighted the underuse of urinary PCR which was not the main point to be retained.

The conclusion has therefore been modified as suggested by the two reviewers:

“This study highlights some advantages of urine PCR in supporting leptospirosis diagnosis during the first week of illness, contrary to prevailing assumptions. Although limited by the small number of concomitant blood and urine samples,

our findings provide clinically relevant insights for optimizing diagnostic strategies. Additional prospective studies where sequential enrollees with febrile illness are tested with all referenced tests are needed to affirm the conclusions suggested by this study. Meanwhile, we suggest that urine PCR should be considered at the first consultation, alongside blood PCR."

Reviewer #3 (Comments for the Author):

Major Strengths

- **Clinical Relevance:** The study addresses a challenging topic for clinicians and provides guidance that could lead to earlier diagnosis and, consequently, better patient outcomes.
- **Novel Findings:** The findings regarding the high sensitivity of urine PCR in the first week of illness are surprising and contradict some existing guidelines. These results directly emphasize the need to re-evaluate diagnostic protocols.
- **Robust Methodology:** The strong statistical analysis, utilizing multivariate logistic regression, correctly identifies factors associated with the sensitivity of blood PCR.
- **Concomitant Sampling:** The evaluation of combined sensitivity in patients who underwent both tests clearly demonstrates the complementarity of the two methods.

Key Points for Revision and Clarification

1. **Study Design:** In the "Methods" section, the authors use the term "cross-sectional study" but describe it as an "observational retrospective study" in the "Ethics" section. This needs clarification to avoid confusion. Many of the study's weaknesses are also due to its retrospective nature, which must be addressed more explicitly in the "Limitations" section.

The methodology was clarified in the methodology section LINE 55-56

"We conducted a retrospective, multicentric observational study in the three main hospitals of French Guiana."

The limitations inherent to retrospective analysis were highlighted in the discussion as follows LINE 336:

"Beyond the inherent limitations of the retrospective design, this study has several weaknesses."

2. **Case Definitions:** The diagnostic criteria for "probable cases," which are defined based on a positive IgM alone without a potential alternative diagnosis, may be open

to criticism. Given the retrospective nature of the study, this is a strong assumption that should be discussed with more detail.

The decision to include patients with IgM alone as a potential criterion for probable cases is indeed debatable. That is why we used an additional criterion to limit the rate of false positive. The absence of a proven differential diagnosis upon review of the files was necessary to consider these cases with only positive IgM.

This element is presented in the study methodology **Line 88-91**:

“Patients with positive anti-Leptospira IgM above 50 IU/ml alone (without positive MAT or PCR) were considered probable cases only if no differential diagnosis was found in the medical records, whether seroconversion of IgM occurred or not. Thus, patients with positive IgM antibodies alone (without MAT or PCR criteria) and with a differential diagnosis were excluded”

The discussion was enriched by the reviewer's comment highlighting this aspect as follows **LINE 336-349**

“First, only patients with robust criteria for biological diagnosis of leptospirosis performed in routine were included. Some clinical guidelines consider an isolated positive IgM sufficient for a probable leptospirosis diagnosis. Yet, we acknowledged the potential lack of specificity of IgM, particularly due to cross-reactions that may introduce classification bias. To reduce false positives, patients with elevated IgM as the sole diagnostic criterion were excluded if an alternative diagnosis was present. Because the study included only patients with confirmed or probable leptospirosis the specificity of urine and blood PCR test was not ascertainable in our study. High sensitivity and specificity of the PCR tests used in this study have been reported.(15, 16) Over 20 years ago, Smythe et al. noted that some discrepancies between negative MAT and positive PCR actually reflected early diagnoses detected by PCR but missed by MAT, rather than true PCR false positives; which was confirmed in many studies since then.(4) Although false positives can occur with any PCR assay, they are rare with qPCR and are unlikely to account for the results observed. Finally, the absence of patients with other diagnosis than leptospirosis was not considered as a major issue for the study objectives.”

3. Sampling Bias: The authors note that urine PCR was prescribed less frequently than blood PCR. While this is an important finding, it could indicate a selection bias, meaning clinicians might have only ordered urine PCR for patients with specific

symptoms or in a particular clinical phase. The potential for this bias should be discussed more thoroughly in the "Discussion" section.

This point was also raised by the previous reviewer and was considered for modification in the discussion as follows **LINE 361-366:**

“Fourthly, it is possible that bias was introduced when clinicians selected the type of clinical samples for leptospirosis diagnosis. This is suggested by the low proportion of negative PCR results among patients who underwent both blood and urine testing. In addition, the higher rate of antibiotic use prior to urine PCR than blood PCR may also indicate clinical differences between patients who did or did not undergo urine testing. This may reflect a selection bias, with urine PCR performed preferentially in patients considered more likely to be infected. ”

4. Influence of Antibiotics: The study shows that antibiotic use before sampling was twice as frequent in the urine PCR group compared to the blood PCR group. While the authors included this factor in their multivariate analysis, this significant difference could confound the results and requires further discussion on its potential implications.

This remark on the difference of antibiotic use in the subjects with urine or blood PCR is relevant and could reflect clinical differences between patients as discussed just above.

Besides, the frequency of empirical antibiotic therapy prior to testing was indeed higher in patients tested by urinary PCR, and the severity of the patients was greater in the group that underwent both tests.

We modified the discussion to comment further the role of antibiotics on PCR positivity **LINE 315-325**

“The analysis of the effect of antibiotic therapy initiation prior to sampling on PCR results was also relevant. First, antibiotic therapy was more frequently administered before sampling for urine PCR than for blood PCR which could indicate a selection bias as commented further in the study limitations. Furthermore, there was a negative association between antibiotic intake before sampling and blood PCR sensitivity in the univariate analysis, in line with previous statements on the negative impact of antibiotic intake on blood PCR (26). Nevertheless, after adjusting for time, this association was no longer significant in the multivariable analysis. Interestingly, there was no association between antibiotic intake and urine sensitivity. Indeed, leptospires are highly sensitive to antibiotics, and their concentrations in blood may be lower than in

urine depending on the stage of the disease, especially in non-severe cases. Thus, although antibiotic intake reduces the urinary excretion of live leptospire (27, 28), urine PCR may still be positive in patients who have been receiving antibiotics for several days. "

5. Data Presentation and Transparency: In the "Results" section, it is mentioned that among 30 patients with concomitant sampling in the first week, 8 had a negative blood PCR but a positive urine PCR. This is a very important finding, but the total number of concomitant samples and their distribution by day are limited, and this limitation should be clearly stated in the conclusion.

As mentioned in other responses to reviewers, we have revised the conclusion to indicate that the number of patients with concomitant samples was limited in this study. Both discussion and conclusion have been revised accordingly:

Discussion **LINE 356-360**

"Thirdly, the number of patients with concomitant blood and urine PCR was limited, as well as their distribution by day. Therefore, it was not possible to conduct the logistic regression analysis of the factors associated with the positivity in this specific population. To corroborate the study findings further works should focus on a larger subset of patients with concomitant blood and urine PCR testing during the first week.

Conclusion

"This study highlights some advantages of urine PCR in supporting leptospirosis diagnosis during the first week of illness, contrary to prevailing assumptions. Although limited by the small number of concomitant blood and urine samples, our findings provide clinically relevant insights for optimizing diagnostic strategies. Additional prospective studies where sequential enrollees with febrile illness are tested with all referenced tests are needed to affirm the conclusions suggested by this study. Meanwhile, we suggest that urine PCR should be considered at the first consultation, alongside blood PCR."

Final Recommendation

This manuscript has high potential for publication due to its clinical importance and novel findings. I recommend that the authors perform a Minor Revision. They must address the points raised above clearly and comprehensively, especially in the "Discussion" and "Limitations" sections, to strengthen the study's validity. After these revisions are made, the article would be well-suited for consideration for publication.

We thoroughly examined the reviewers' comments. This led to clarify the limitations of the study emphasizing the limited number of patients with concomitant samples, the retrospective nature of the study, and the absence of standardized sampling, particularly in relation to MAT, while noting the limitations of MAT as the gold standard for early samples.

The Bullet Points should be modified as follows:

- **The optimal timing of urine PCR for leptospirosis diagnosis is controversial**
- **Among 188 patients diagnosed with leptospirosis in French Guiana, urine PCR was performed in half as many patients as blood PCR**
- **Most urine PCR tests were done in the first week of illness with a high positivity rate**
- **A longer delay between symptoms onset and sampling was associated with negative results for blood PCR but not for urine PCR**
- **Urine PCR should be considered at the first consultation, alongside blood PCR**

Re: Spectrum01185-25R1 (Urine PCR testing as an effective method for early diagnosis of leptospirosis)

Dear Dr. Paul Le Turnier:

Your manuscript has been accepted, and I am forwarding it to the ASM production staff for publication. Your paper will first be checked to make sure all elements meet the technical requirements. ASM staff will contact you if anything needs to be revised before copyediting and production can begin. Otherwise, you will be notified when your proofs are ready to be viewed.

Sincerely,
Mark Pandori
Editor
Microbiology Spectrum